# MEASURING IN-CONTEXT ABILITY OF STEERED REPRESENTATION IN LARGE LANGUAGE MODELS

## ABSTRACT

Large Language Models (LLMs) show advanced performance and adaptability across various tasks. As the model size becomes more extensive, precise control by editing the forward process of LLMs is a challenging problem. Recent research has focused on steering hidden representations during forward propagation to guide model outputs in desired directions, yielding precise control over specific responses. Although steering shows a broader impact on diverse tasks, the influence of steered representations remains unclear. For instance, steering towards a refusal direction might lead the model to refuse even benign requests in subsequent generations. This work tackles the problem of evaluating activation steering. We introduce a counterfactual-based steering evaluation framework that compares the output of base and steered generations. Within the framework, we propose a steering effect matrix that eases the selection of generations base and steered output types. We experimentally evaluate the effects of steered representation for consequence generation with Llama3-8B, Llama2-7B, and Exaone-8B across diverse datasets. We conclude that steered representation changes the original output severely in longer contexts.

## 1 INTRODUCTION

The transformer architecture has demonstrated high performance in integrating context information to generate output (Vaswani et al., 2023). The ability to process in-context information is known to naturally develop during the training process of transformer models that predict the next word in a sequence (Brown et al., 2020; Chen et al., 2022b). To control transformer models, methods such as parameter fine-tuning and improving model instructions have been continuously attempted (Ouyang et al., 2022; Bai et al., 2022).

Recent research has focused on activation steering, which involves directly modifying the hidden representations during the forward pass to guide the output in the desired direction. Steering has shown effectiveness in altering the model output, proving that it can serve as a method for controlling the decoding steps (Liu et al., 2024; Turner et al., 2024; Niu et al., 2024; Luo et al., 2024). Since it can influence the overall output of the model in a specific direction, it is expected to contribute to controlling the behavior of the model according to the desired characteristics, including safety considerations (Arditi et al., 2024; Zheng et al., 2024a; Turner et al., 2024).

However, most of the existing research on steering has not verified whether the steered generation follows human instruction (e.g., the format of the answer) or has not assessed the effects of steering in longer contexts. In addition, although there are various studies on making steering more effective, the evaluation of consequence generation across different concepts is not conducted. As steering methods are expected to play an important role in controlling models in the future, there is a need for a framework to analyze their influence. In this paper, we tackle the problem of evaluating the contribution of steered representation to consequence prompts. Figure 1 shows the overall evaluation framework.

We suggest (1) the format-preserving rate (FPR), whether the activation steering can preserve the output formation, (2) the steering success rate (SSR), whether the activation steering encourages the output to the positive (intended) direction, and (3) the side-effect rate (SER), whether the activation steering causes side effects. To evaluate this, we also propose a steering evaluation matrix, which is motivated by a confusion matrix, that counts base generation cases and steered output cases so that

Figure 1: Evaluation protocol of steered generation. After the completion of the steered output, additional prompts (relevant or irrelevant) can be provided. The effects of the steering on the generation are evaluated compared to the base generation (without steering).

we can count how many cases are successful. Based on the evaluation framework, we explore the effects of a steered representation (on a single token) on the generation of consequences. We ask the following questions:

- Can steering increase the concept of output while maintaining formatting?
- How can we calculate unexpected steered outputs?
- Can a single token steering activation contribute to the generation of the intended direction?
- Are prompts for irrelevant tasks also influenced in the intended direction by the keys and values of the steered representation?

In conclusion, this work provides an extensive view of activation steering evaluation. This work can contribute to more reliable and safe activation steering for various purposes.

## 2 PRELIMINARY

In this section, we provide background on activation steering and in-context learning of LLMs and describe the notations used in this work.

**Activation Steering** Activation steering refers to the process of modifying a model's hidden state to control its output (Niu et al., 2024; Turner et al., 2024). In LLMs, steering is applied for various purposes such as debiasing undesired toxic and harmful requests (Arditi et al., 2024), and controlling text styles (Liu et al., 2024; Konen et al., 2024). The most common steering procedure is to (1) compute the conceptual direction $r$ from binary labeled samples and (2) modify a hidden representation $h$ with a given magnitude control parameter $\alpha$ by $h \to h + \alpha \cdot r$ in the forward pass. Activation steering considers several configurations, such as token locations, target layers, transformer modules, and methods for obtaining conceptual directions. Although activation steering successfully encourages the desired behavior of LLMs, it is non-trivial how the steered representation affects the consequence generation of LLMs. This work conjectures that unexpected side effects of activation steering exist.

**In-context Learning** We briefly describe the backbone model, transformer architecture (Vaswani et al., 2023). A transformer decoder block has multi-head attention (MHA), which transports key and value representations to a query with causal masking, and a multi-layer perceptron (MLP), which has two linear layers with nonlinear activation. In-context learning of LLMs refers to the ability to store and retrieve task-relevant in-context information during the inference phases (Brown et al., 2020). This allows better generalization for out-of-distribution samples for a given task and superior performance on unseen tasks. Recent studies focus on finding special attention heads (Zheng et al., 2024b) such as the previous token head (Elhage et al., 2021), the induction head (Olsson et al., 2022), and an attention head for question answering biased option selection (Burns et al., 2023). Furthermore, recent studies show that LLMs, when provided with in-context demonstrations, encode task-specific (Hendel et al., 2023) or function-specific (Todd et al., 2024) hidden representations, which play a pivotal role in the process of in-context learning.

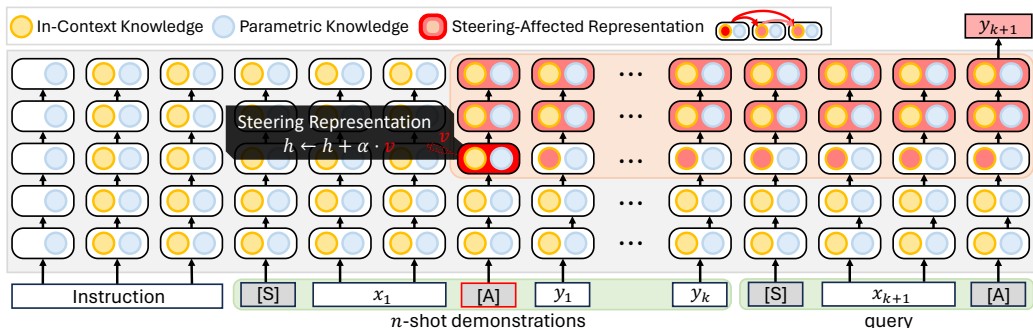

Figure 2: A template of few-shot prompting. The prompt comprises instruction, $k$-shot demonstrations, and a query $x_{k+1}$ in sequence. Regardless of the number of shots $k$, we steer at the location of the symbol [A] after the first example. The steered direction (highlighted in red) influences the hidden representation following the steered layer, resulting in the output $y_{k+1}$.

## 2.1 NOTATION

We consider input and output paired dataset $\mathcal{D} = \{(x_i, y_i)\}$ with input text $x_i$ and binary label $y_i \in \{y_{\mathrm{neg}}, y_{\mathrm{pos}}\}$. For the task description, we have instruction $I$ and denote $E_k = (x_1, y_1, x_2, y_2, \cdots, x_k, y_k)$ as few-shot examples where $(x_i, y_i)$ is a uniformly sampled example. We construct a $k$-shot prompt by concatenating the instruction $I$, few-shot examples $E_k$, and query $x_{k+1}$. We add special tokens [S] and [A] before the input $x$ and the label $y$, respectively. Figure 2 shows the template used for the few-shot prompting. We steer only a **single** hidden representation of the [A] token location after the $x_1$. In a zero-shot setting, the steered token location matches the query token [A], affecting the generation of the next token. In a $k$-shot ($k > 0$) setting, the steered representation is key and value, so the query token attends to the steered representation.

Transformer decoder is an autoregressive model that computes likelihood $p_\theta(z_t | z_{1:t-1})$ of a next token $z_t$ given previous tokens $z_{1:t-1}$. For a given text $z$, we evaluate whether $z$ matches a positive label with an indicator function

$$\mathbb{1}_{\mathrm{pos}}(z) = \begin{cases} 1 & z = y_{\mathrm{pos}} \\ 0 & \text{otherwise.} \end{cases} \tag{1}$$

We also define $\mathbb{1}_{\mathrm{neg}}(z)$ similarly for the negative label $y_{\mathrm{neg}}$. For the generated text, we denote base generation by $z^{\mathrm{base}} \sim p(z)$ without steering and $z^{\mathrm{steer}}$ for the steered generation.

## 3 MEASURING EFFECTS OF STEERING

In this section, we define a simple yes/no answering task and evaluate how activation steering affects generation performance. We define two types of evaluation metrics, one for task **format-preserving rate** and the other for **steering success rate**. Some evaluations (Zhang & Nanda, 2024) of activation patching utilize logit-based evaluation. However, we consider token-based evaluation because (1) the generation length could be longer than a single token and (2) the target token to compute logit is vague.[1]

## 3.1 YES/NO TOKEN GENERATION TASK

Steering is a treatment for increasing the conceptual meaning of the generation, which might require different evaluation metrics. In this work, we define a yes/no generation task, where the instruction to guide LLMs to generate "`yes`" token for the positive and "`no`" for negative label cases. After the instruction, few-shot examples can be provided. For steering, we fix the steering location to the first [A] token.

---

[1]The tokenization of the target word could differ depending on the previous characters. For example, `X:` `yes` could be tokenized by without space ("`X: `", "`yes`") or with space ("`X:`", "` yes`").

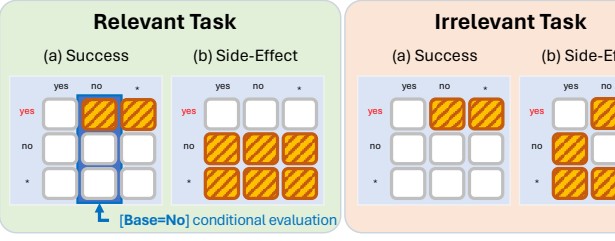

Figure 3: Steering evaluation matrix whose entry represents the number of samples of steering results(row and column). The left matrix shows the locations for the base and steered outputs, and the first row (colored red) represents the target steered label. The symbol $\star$ indicates other tokens rather than label tokens (yes and no). The right four matrices show the success and side-effect cases for relevant and irrelevant tasks, respectively. The dashed entries represent the targeted cases for each measure. The normalization term can be chosen for the condition-specific measures.

## 3.2 MEASURING FORMAT-PRESERVING RATE (FPR)

First, we evaluate how much the steered output can preserve the original task form. We define the format-preserving rate (FPR) by

$$\text{Format}^{\star} = \Big( \sum_{z^{\star} \in \mathcal{D}} \mathbb{1}_{\text{pos}}(z^{\star}) + \mathbb{1}_{\text{neg}}(z^{\star}) \Big) / |\mathcal{D}| \tag{2}$$

where the symbol ($\star$) is either base or steered setting. We measure how much the generation format is preserved after the steering by

$$\delta_{\text{format}} = \text{Format}^{\text{steer}} - \text{Format}^{\text{base}} . \tag{3}$$

This measure is expected in range (-1,1), and zero is when the steered output has the same proportion of data preserving the format. In general, steering modifies the original output, and the task rule, such as the output format, might not be preserved. Therefore, the expected sign of $\delta_{\text{format}}$ is negative.

## 3.3 MEASURING STEERING SUCCESS RATE (SSR) AND SIDE-EFFECT RATE (SER)

We consider three types of outputs: yes, no, and $\star$. The symbol $\star$ is the case when the generation is neither yes nor no. We evaluate the steering by measuring the amount of variation across these three values. Note that comparing pre- and post-steering resembles counterfactual framework Vig et al. (2020). For the evaluation of pairs, we propose $3 \times 3$ steering evaluation matrix $\mathcal{S}$ whose entries represent the base generation result (source) to the steered generation result (target). We define a general ratio by

$$\text{Rate}_{\mathcal{N}}^{\mathcal{T}} = \frac{\sum_{(i,j) \in \mathcal{T}} S_{i,j}}{\sum_{(i,j) \in \mathcal{N}} S_{i,j}} \tag{4}$$

where $\mathcal{N}$ is a set of entries for normalization and $\mathcal{T}$ is a set of targeted entries with condition $\mathcal{T} \subseteq \mathcal{N}$. The choice of $\mathcal{T}$ and $\mathcal{N}$ provides the targeted quantity. In this work, we consider three kinds of $\mathcal{T}$:

- **Steering Success Rate** is when the the non-yes label is steered to yes label. Therefore, we construct target set $\mathcal{T}_{\text{succ}} = \{S_{n \to y}, S_{\star \to y}\}$.
- **Steering Side-Effect Rate in the Relevant Task** is when the positive label is not achieved. $\mathcal{T}_{\text{rel.fail}} = \{S_{y \to n}, S_{y \to y}, S_{n \to n}, S_{n \to \star}, S_{\star \to n}, S_{\star \to \star}\}$.
- **Side-Effect Rate in the Irrelevant Task** is when the original category is converted to another category; therefore, $\mathcal{T}_{\text{irr.fail}} = \{S_{n \to y}, S_{n \to \star}, S_{\star \to y}, S_{\star \to n}, S_{y \to \star}, S_{y \to n}\}$

We use terms **steering success rate (SSR)** to indicate the proportion of $\mathcal{T}_{\text{succ}}$ and **side-effect rate (SER)** to indicate the proportion of $\mathcal{T}_{\text{rel.fail}}$ or $\mathcal{T}_{\text{irr.fail}}$. One natural choice for the normalization term $\mathcal{N}$ is counting all samples in $S$. However, we may consider the conditional case when the base generations of no are changed to yes. For this, we use notation **SSR+**. Figure 3 shows three measures' steering evaluation matrix, evaluation cases, and normalization conditions. We discuss the semantic meaning of the measures and normalization conditions in the Appendix B.1.

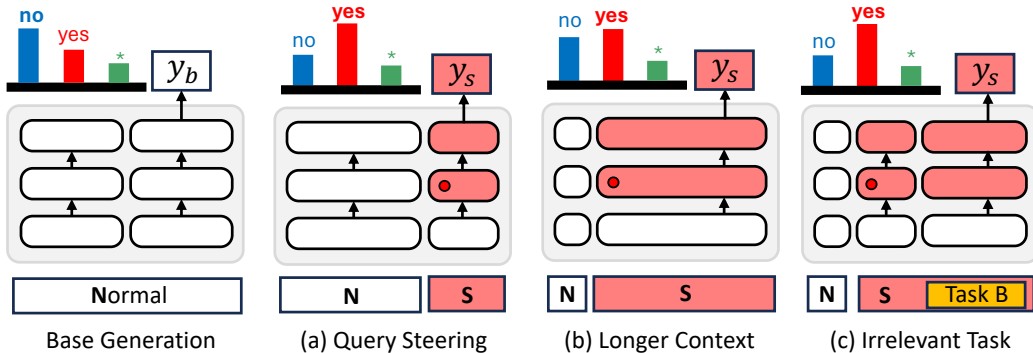

Figure 4: Steering effects cases. We refer to the phase before steering as $N$ (base), and the phase after steering as steered $S$ (steered). Small red dots in layers indicate the steering locations, and the descriptions of each case are as follows. Base Generation: The model generates the output $y_b$ without steering. (a) Query Steering: The model is steered toward *yes* at the query location, generating $y_s$. (b) Longer Context: The model is steered before the query token (c) Irrelevant Task: The model performs a new task, denoted as Task B.

# 4 STEERING EFFECTS ON CONSEQUENCE GENERATION

Transformer decoder generates consequence tokens until the end of the sentence token is generated, and the early steered generation affects the consequence generation. When steering an activation, the original probability $p(z_t|z_{1:t-1})$ is conditional on modified hidden representation rather than tokens $z_{1:t-1}$. With slight abuse of notation, we denote $p(z_t|H_{1:t})$ to represent the conditional probability with respect to hidden representation $H_i \in \mathbb{R}^{L \times d}$ where $L$ is the number of layers and $d$ is hidden dimension. We may consider samples from $z^{\text{steer}} \sim p(\cdot|S_{1:t})$ where $S_{1:t}$ is the steered hidden representation of $H_{1:t}^{1:L}$ in a transformer decoder. When we steer a token at location [A] of layer $\ell$, we have the following conditioning,

$$z^{\text{steer}} \sim p(\cdot|[H_{1:N}; H_{[A]}^{1:\ell}; H_{[A]}^{\ell:L}; H_{S:t}^{1:\ell}; H_{S:t}^{\ell:L}]) \tag{5}$$

where $N$ and $S$ are token locations right before and after [A] (red color represents affected hiddens). The steering-affected hiddens could be out-of-distribution; for example, steering a token-embedding in a random direction is similar to adding random tokens, which is a severe problem in adversarial attacks in LLMs such as jailbreak (Shen et al., 2024). To differentiate the cases of steering-affected hiddens, we consider the following cases:

1. **Query token**: steering hidden representation at the query token location.

2. **Consequence relevant task**: One of the previous tokens is steered, and additional prompts for the relevant task are provided.

3. **Consequence irrelevant task**: One of the previous tokens is steered, and additional prompts for the irrelevant task are provided.

Figure 4 shows the base generation and three steered cases. We evaluate the proposed measures for each case and reveal how the steering affects query-location and consequence generations.

# 5 EXPERIMENTS

We use datasets Paradetox (Logacheva et al., 2022), SubjQA (Bjerva et al., 2020), and Jailbreak (Shen et al., 2024) datasets to gather steering vectors from a train split and evaluate the effects of steering on the test samples. We construct two-shot and four-shot datasets of sample size 1000. We use the same samples for the base and steered generation. We use Llama3-8B-inst (AI, 2024), Llama2-chat-7B (Touvron et al., 2023) and Exaone-8B (Research et al., 2024). We use greedy decoding for a generation. Following the previous convention of activation steering, we cluster

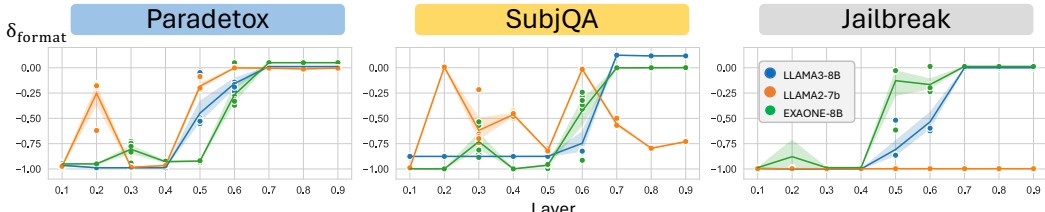

Figure 5: Format-Preserving after steering by model and dataset under a zero-shot setting. The y-axis represents $\delta_{\text{format}}$, where -1 indicates the format is not preserved, 0 means the format is preserved and more than 0 is better preserved. The x-axis corresponds to the transformer layers (0.1 to 0.9). Llama3-8B(blue) and Exaone-8B(green) preserve the format well above layer ratios of 0.7 across datasets. EXAONE-8B performs better in Paradetox, while Llama3-8B excels in SubjQA. Llama2-7B (orange) struggles with format-preservation, particularly in SubjQA and Jailbreak.

positive and negative labels by evaluating samples for generation rather than using the original label (Arditi et al., 2024) and gather hidden representations at residual stream (Geva et al., 2021).

For the irrelevant tasks, we consider simple tasks for answering yes or no formats. We set capital/country (Todd et al., 2024), positive/negative sentiments (Todd et al., 2024), animal/person question types (Talmor et al., 2019), and upper/lower cases (Todd et al., 2024). These datasets are expected to not be affected by steered hidden representations.

We use the most common approach for the steering method, additive steering (Liu et al., 2024). We collect the activation at [A] location for separated positive and negative generations and compute the mean differences (Jorgensen et al., 2023).

$$\bar{a}_{\text{pos}} = \frac{1}{\mathcal{P}_{\text{pos}}} \sum_{y \in \mathcal{P}_{\text{pos}}} h^{\ell}_{[\text{A}]}(y) \tag{6}$$

$$\bar{a}_{\text{neg}} = \frac{1}{\mathcal{P}_{\text{neg}}} \sum_{y \in \mathcal{P}_{\text{neg}}} h^{\ell}_{[\text{A}]}(y) \tag{7}$$

$$\tilde{r} = \frac{\bar{a}_{\text{pos}}}{\|\bar{a}_{\text{pos}}\|_2} - \frac{\bar{a}_{\text{neg}}}{\|\bar{a}_{\text{neg}}\|_2} \tag{8}$$

We steer hidden representation $h$ by

$$\tilde{h} \leftarrow h + \alpha \cdot \tilde{r}/\|\tilde{r}\|_2. \tag{9}$$

Then, we normalize $\tilde{h}$ to preserve the original norm. The steered representation affects only the upper blocks and the subsequent tokens, as visualized in Figure 2.

## 6 RESULTS

### 6.1 FORMAT-PRESERVING AFTER STEERING AND POSITIVE LABEL RATIO

Activation steering is the process of indirectly altering the model's response. Therefore, it is essential to evaluate not only whether the model generates semantically desired answers in the target direction but also whether it adheres to the format specified by the user's instructions. We request the model to respond in a `yes` or `no` format and observe whether the response maintains this format after steering in the zero-shot setting (equivalent to query steering in Figure 4).

Figure 5 presents the results of measuring the Format-Preserving Rate (FPR), as defined in Section 3.2 for $\alpha \in \{0.2, 0.4, 0.6, 0.8, 1.0, 5.0\}$. Intuitively, if the format is preserved after steering, the FPR is close to 0, while a format collapse results in a value close to -1. The results indicate that **activation steering applied to layers above 0.7 percentile effectively preserves the format**. However, steering to layers below the 0.4 percentile leads to significant format disruption. This disruption varies across models and datasets.

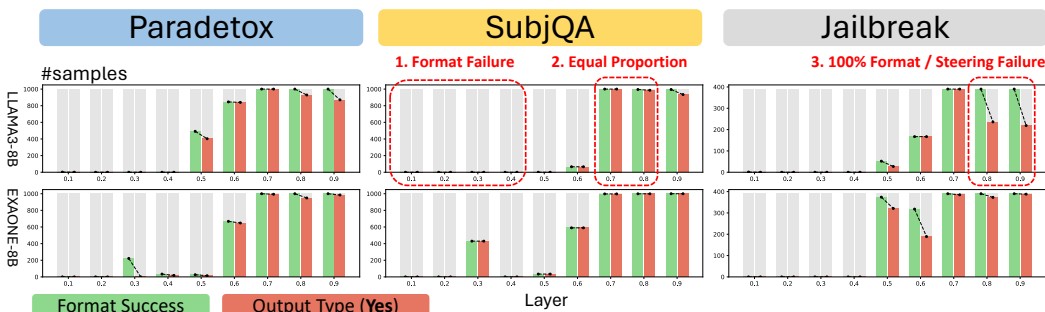

Figure 6: The Positive Label Ratio after steering. We measured the performance across different layers for two models(Llama3-8B and Exaone-8B) on three datasets: Paradetox, SubjQA, and Jailbreak under a zero-shot setting. Gray bars indicate the number of samples, while green and red bars indicate the number of format successes and "Yes" output types, respectively. In the Llama3-8B model on the SubjQA dataset, format failures can be observed in the early layers (0.1 to 0.5), while at layers 0.7 and 0.8, there is an equal proportion of format success and "Yes" outputs. On the Llama3-8B model for the Jailbreak dataset, although the format is fully preserved, the number of "Yes" outputs decreases after steering.

We also evaluate whether the model's generation is effectively steered in the intended (yes) direction while maintaining the requested format. Figure 6 represents the number of samples where the model outputs yes, specifically focusing on examples where the format is preserved after steering. We measure how many samples maintain the format after steering (colored in green) and how many samples are guided toward the yes label (colored in red) compared to the total number of the dataset (colored in gray). In most cases, we observe that the format is conserved while successfully steering to the yes direction. However, format-preserving does not guarantee the success of steering. Notably, as seen in the results of Llama3-8B, the output format remains ideal after steering, but the response is not pulled in the positive direction. **This observation suggests that the model's answer may remain `no` even after steering.** The steering outcome can succeed or fail, regardless of whether the format is preserved.

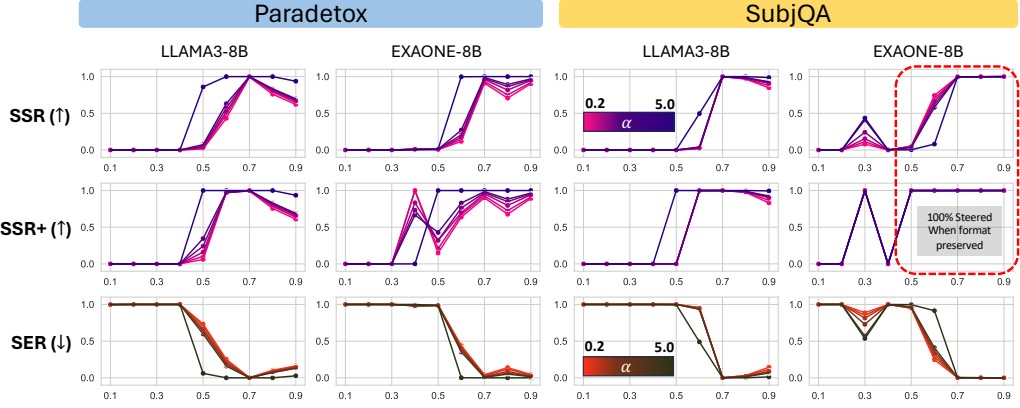

Figure 7: Evaluation on steering effects across different $\alpha$ values and ratio layers under a zero-shot setting. We observe sharp jumps for all measures. A higher SSR and SSR+ indicate better steering performance, while a lower SER signifies fewer side effects of the steering process. In the red dashed area, Exaone-8B achieves 100% steering success when the format is preserved.

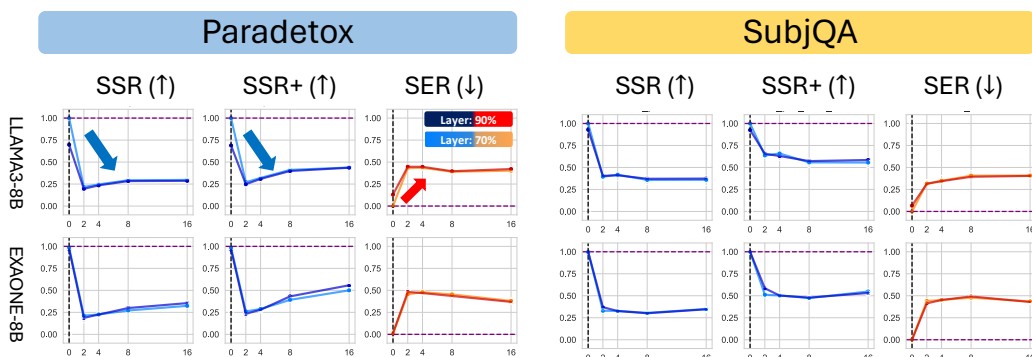

Figure 8: Steering Success Rate(SSR), Format-Preserved Steering Success Rate(SSR+), and Side Effect Rate(SER) across two layers, 0.7 and 0.9. The X-axis represents the increase in shots, and the Y-axis represents the steering success rate. The purple dashed line is achieved when the steering effect remains even for the longer-context example. In the Paradetox dataset, Llama3-8B shows a sharp decline in SSR and SSR+ at 2 shots, followed by a gradual increase, leveling off around 8 shots with similar results. For SER, Llama3-8B rises at 2 shots, then decreases and levels off afterward. These results indicate that the steered effects do not remain high after additional prompts.

## 6.2 QUERY SUCCESS RATE

Figure 7 presents the SSR and SER measurements for the model's output after steering. Steering Success Rate (SSR) is calculated across all samples, while SSR+ is normalized for format-preserving examples. Higher values for both metrics indicate more successful steering in the desired direction. In most settings, we observe that as alpha increases and steering is applied to higher layers, the steering success rate improves.

However, Some exceptions still exist. For instance, in the SubjQA dataset, Exaone-8B shows a decrease in SSR as the alpha value increases. Additionally, SSR is higher at the 0.3 layer depth than at the 0.4 layer depth. Another point is that although SSR appears relatively low at the 0.6 layer depth, steering is highly effective for samples where the format is well-preserved. That is, the analysis of steering success can vary depending on the criteria used for normalization. Side Effect Rate (SER) increases when the model fails to generate outputs in the desired direction. In most cases, SER is higher in the lower layers, indicating that labels are often formed as something other than 'yes'. These findings demonstrate the need for a more refined evaluation method that takes into account a wider range of conditions when assessing steering performance.

## 6.3 LONGER EFFECTS ON THE SAME TASKS

We explore how the effects of steering change as the distance from the steering location increases. Due to the localized token interactions in LLMs (Chang et al., 2024), the steering effect may diminish as the distance grows. However, given the nature of decoder-based models, which build representations based on causal modeling, predicting how steering will affect more distant tokens is non-trivial. Therefore, we measure the effect of steering as the context length increases for few-shot examples. We steer at the location of the first example, 70% layer, $\alpha = 1$, and evaluate whether the $k$-th generation is modified compared to the original greedy decoding output. Figure 8 shows the evaluation results in a $k$-shot setting. We observe that the SSR performance significantly drops below 0.5 compared to query steering performance. **As SSR drops to lower and SER becomes larger than 0, the in-context ability of steered representation is weak.**

## 6.4 LONGER EFFECTS ON IRRELEVANT TASKS

Unlike expecting the in-context ability of steered representation for relevant tasks, the steered representation (yes-direction) should not influence the base generation output. We evaluate the outputs of irrelevant tasks by two means: (1) label-yes direction should not be presented after steering task

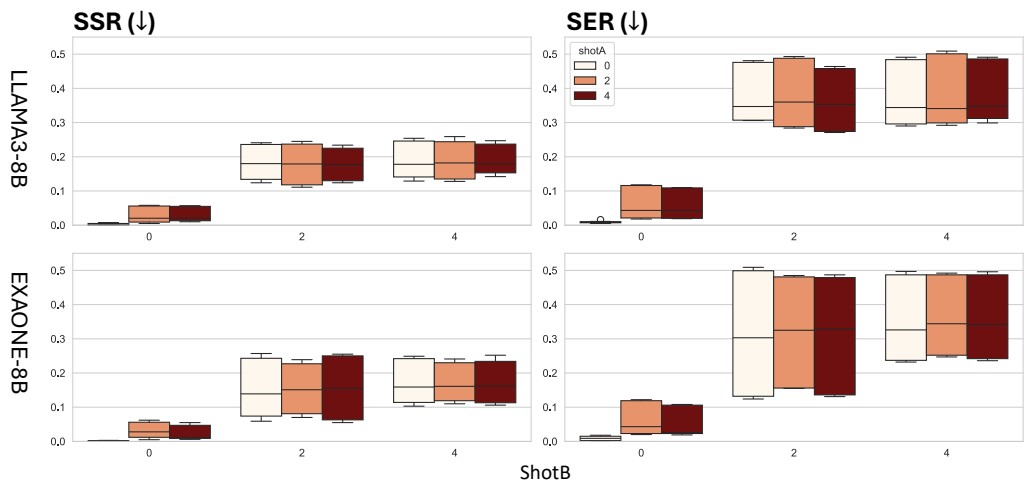

Figure 9: SSR and SER rates for irrelevant tasks. The X-axis is the number of shots provided for task B. For each case, we extend the context even longer for task A's increased number of shots. We observe the increase of two measures. The samples for each box plot are different irrelevant datasets. As the number of shots increases, the results indicate output label conversion.

A (low ($\mathcal{T}_{\text{succ}}$)), and (2) any label should not be modified (low ($\mathcal{T}_{\text{irr.fail}}$)). Figure 9 shows two measures for four irrelevant datasets depending on the context length. As the context length increases, the proportion of yes increases and more labels are modified. These results indicate that **single steering at the early part modifies the long-context output.**

## 7 RELATED WORK

**In-context Learning** In-context learning (ICL) has emerged as a powerful paradigm in natural language processing, enabling large language models (LLMs) to adapt to new tasks without parameter updates (Brown et al., 2020; von Oswald et al., 2023; Dong et al., 2024). With only a few examples, ICL can effectively integrate human knowledge into LLMs and it also offers the advantage of generating responses in the desired format. Hendel et al. (2023) and Todd et al. (2024) describe the learning process of ICL utilizing the concept of a *Task (or Function) Vector*. They suggest that ICL compresses the demonstration set into the vector, which is used to generate outputs. Todd et al. (2024) and Olsson et al. (2022) propose that an attention mechanism known as *Induction Heads* is responsible for most of the in-context learning, explaining the underlying principles of ICL.

Despite its advantages, ICL has several side effects. Users can manipulate prompts to induce the model to generate harmful responses (Shen et al., 2024; Zou et al., 2023b; Anil et al., 2024). Furthermore, Min et al. (2022) and Liu et al. (2021) observe that as the context length increases, the model struggles to learn the task. Chen et al. (2022a) and Xie et al. (2024) find that LLMs are highly receptive to external knowledge, such as context, rather than relying on memory knowledge.

**Mechanistic Interpretability** Recent research on Mechanistic Interpretability (Nanda & Bloom, 2022; Geiger et al., 2021; Elhage et al., 2021; Geva et al., 2023; Xie et al., 2024; He et al., 2024; Merullo et al., 2024; Lee et al., 2024a) attempts to interpret LLMs by examining their internal components, such as neurons, circuits, attention heads, and multi-layer perceptrons. The output of mechanical units, known as activations, is formed as a linear combination of multiple neurons and serves as a key tool in analysis (Zou et al., 2023a; Zhang & Nanda, 2024; Park et al., 2024). Additionally, the direction of the activation encodes information about the concepts within the model (Burns et al., 2023; Li et al., 2023). Expanding on the understanding, the possibility of manipulating models by steering their internal activations has attracted significant interest.

**Activation Steering** Activation Steering refers to a set of modification techniques that adjust the direction of activations to produce the desired output of LLMs. The target direction can be derived from data samples (Jorgensen et al., 2023; Rimsky et al., 2024), or identified through dictionary

learning methods (Cunningham et al., 2023; Marks et al., 2024), such as codebook. We focus on the former approach, which extracts the direction using only a few text examples. Niu et al. (2024) introduced an approach for adapting models using only the forward pass without backpropagation, which inspired the steering methods. Liu et al. (2024) and Turner et al. (2024) eliminate the need for demonstration selection by injecting context information as a vector, making the model more controllable. Wu et al. (2024) defines a Low-rank Linear Subspace and explores the model's internal causality through activation engineering. Lee et al. (2024b) proposes Conditional Activation Steering (CAST), demonstrating that the outputs of LLMs can be selectively adjusted based on the input context.

**Safety Alignment** The primary goal of safety alignment is to ensure that LLMs follow user instructions while rejecting harmful or unethical requests. Unlike previous works (Wang et al., 2024a; Stickland et al., 2024; Shen et al., 2024; Zou et al., 2023b; Lee et al., 2024a) that focused on the perspective of prompt engineering, recent research aims to achieve safety alignment by applying steering approaches. Arditi et al. (2024) et al. reveal that the refusal mechanism in LLMs can be mediated by a single-direction vector. By leveraging this, they propose a new method to jailbreak LLMs, highlighting the instability of current safety alignment methods. Zheng et al. (2024a) and Turner et al. (2024) propose a new framework to enhance LLM safety by adopting an approach that manipulates activations.

# 8 DISCUSSION

The instability caused by steering can become severe for the long generation of transformer decoders. For example, we may not know whether steering a single representation in 1M tokens can cause noisy outputs. To evaluate the effect of steered representations, we explored the use of steering vectors to adjust the generative direction of large language models (LLMs) by applying them to hidden states. The results show that steering vectors can be a powerful tool for guiding the model's output in a desired direction (Wang et al., 2024b). However, a more refined understanding of how to generate and apply steering vectors is necessary for precise control.

One of the key findings is that steering vectors can affect more than simply steering the model's output toward affirmative responses. For example, we observed that steering vectors could influence response formats or even impact tasks outside the intended scope (Logacheva et al., 2022; Stickland et al., 2024). This suggests that steering vectors may have more complex and diverse effects, indicating the need for further investigation into their broader impact (Niu et al., 2024; Turner et al., 2024).

We found that the optimal steering location and effects varied between models and datasets, making it difficult to establish a predictable output after steering representation Tan et al. (2024). In conclusion, while steering vectors offer significant potential for guiding LLM behavior, the research on their precise application and measurement is still incomplete. Establishing standardized frameworks and evaluation metrics will be essential to fully harness the potential of steering vectors while minimizing unintended side effects.

# 9 CONCLUSION

In this work, we evaluate the effect of a single token activation steering for consequence generation. For this purpose, we propose a steering effect matrix to consider the choice of output types and construct two measures: steering success rate and side effect rate. Experimental results show that steering at a query location can motivate the desired output. For longer relevant tasks, the in-context ability of steered representation is weakly presented. Lastly, for the irrelevant tasks, the steered representation modified the original output, indicating the existence of side effects. Overall, we evaluate the effects of steered representation and show the existence of instability caused by steering. Therefore, steering methods that consider these effects must be studied to control the generation's controllability with activation steering.

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
