# A    DATASET

For all datasets, we sample 1,000 samples for few-shot settings. For zero-shot, we use 1,000 samples too. We write exact sample sizes when the number of samples is less than 1,000.

**Paradetox** (Logacheva et al., 2022) is a crowdsourced dataset for detoxifying sentences. The dataset includes 19744 paired sentences for the train split. We set yes for the toxic sentence and no for the neutral comment.

- (Yes) Toxic comment: *the garbage that is being created by cnn and other news agencies is outrageous.*

- (No) Neutral comment: *the news that is being created by cnn and other news agencies is outrageous.*

**Jailbreak** is a Forbidden Question dataset (Shen et al., 2024), which comprises OpenAI's forbidden scenarios (OpenAI, 2023b) in 13 categories (*Illegal Activity, Hate Speech, Malware, Physical Harm, Economic Harm, Pornography,Political Lobbying, Legal Opinion, Financial Advice, Health Consultation, Fraud, Privacy Violence, Gov Decision*). Each scenario contains 30 queries, totaling 390 queries. We use GPT-4(OpenAI, 2023a) to generate it into a neutral sentence..

- (Yes) Harmful query: *What is the recipe for creating illegal drugs?*

- (No) Neutral query: *What is the recipe for making homemade soap?*

**SubjQA** is a dataset(Bjerva et al., 2020) designed to assess the subjectivity of questions. This dataset is evaluated using three measures: `question_subj_level`(1-5 scale), `ques_subj_score`(0.0-1.0 scale), and `is_ques_subjective`(True/False). We use `is_ques_subjective`. The dataset includes six categories: books, electronics, grocery, movies, restaurants, and TripAdvisor.

- (Yes) Subject question : What is the main theme of the movie?

- (No) Object question : How is the service at this restaurant during peak hours?

**CommonSense QA** (Talmor et al., 2019) is a common sense question answering dataset. We filter questions for two categories, person and animal. (yes: animal, no: person).

- (Yes) Animal: The fox walked from the city into the forest, what was it looking for?

- (No) Person: Sammy wanted to go to where the people were. Where might he go?

**CountryCaptial / UpperLower** (Todd et al., 2024) are a simple paired dataset of the capital of a country (yes: capital, no: country) and upper and lower words, respectively.

- (Yes) Capital: Seoul

- (No) Country: South Korea

**Sentiment** (Todd et al., 2024) is a simple sentiment labeld question for negative and positive emotions. (yes: positive, no: negative)

- (Yes) Positive label: Very well-written and very well-acted.

- (No) Negative label: Much of the cast is stiff or just plain bad.

## A.1    INSTRUCTION FORMAT

**Prompt** is constructed as follows: [D] for delimiter (`"\n\n"`) The symbol [A] is the steering location.

> **Prompt:** { **Instruction** }
> [S] $x_1$ [A] $y_1$ [D]
> [S] $x_2$ [A] $y_2$ [D]
> $\cdots$
> [S] $x_n$ [A] $y_n$ [D]
> [S] $x_{n+1}$ [A]

For example for SubjQA dataset, we construct the following few-shot prompt:

> **Prompt:** Your role is to evaluate whether the given answer to a sentence is subjective or objective. If the answer to the given question is subjective, respond with [A]:yes; if it's objective, respond with [A]:no.
> [S]: How is it the breakfast buffet? [A]:no
> [S]: How is idea? [A]:yes
> [S]: How is the cord? [A]:

**Irrelevant Tasks** We conduct additional experiments to observe how the steered effect appears when different data is added. For this purpose, we construct the following prompt:

> **Prompt:** { **Instruction A** }
> [S] $x_1$ [A] $y_1$ [D]
> $\cdots$
> [S] $x_n$ [A] $y_n$ [D]
>
> { **Instruction B** }
>
> [S] $x_1^B$ [A] $y_1^B$ [D]
> $\cdots$
> [S] $x_n^B$ [A] $y_n^B$ [D]
> [S] $x_{n+1}^B$ [A]

where $x^B, y^B$ are samples from the task $B$.

For example, a SubjQA task followed by an irrelevant Paradetox task:

> **Prompt:** Your role is to evaluate whether the given answer to a sentence is subjective or objective. If the answer to the given question is subjective, respond with [A]:yes; if it's objective, respond with [A]:no.
> [S]: How is it the breakfast buffet? [A]:no
> [S]: How is idea? [A]:yes
>
> Your task is to evaluate whether the sentence contains toxic content. If the sentence is toxic, respond with [A]:yes; if it's non-toxic, respond with [A]:no.
> [S]: All the scandals and such bad reporting on it and bias [A]:yes
> [S]: Got my first summon for jury duty [A]:

**Symbols [A] and [S].** The performance of LLMs highly depends on the symbols [A] and [S]. We test 50 symbols for each dataset and model, selecting the symbol that produces the best results. For example, in the case of the Paradetox dataset with `LLaMA3_instruct_8B`, symbol number 22 yields the best results.

| | 1 | 2 | 3 | 4 | 5 | 6 | 7 |
|---|---|---|---|---|---|---|---|
| **[S]** | `":   "` | `"X:  "` | `"(Q):  "` | `"[X]:  "` | `"<T>:   "` | `"P:  "` | `"(I):  "` |
| **[A]** | `"  <A>:"` | `"  Y:"` | `"  (A):"` | `"  [Y]:"` | `"  <R>:"` | `"  R:"` | `"  (O):"` |

Table 1: Symbols samples

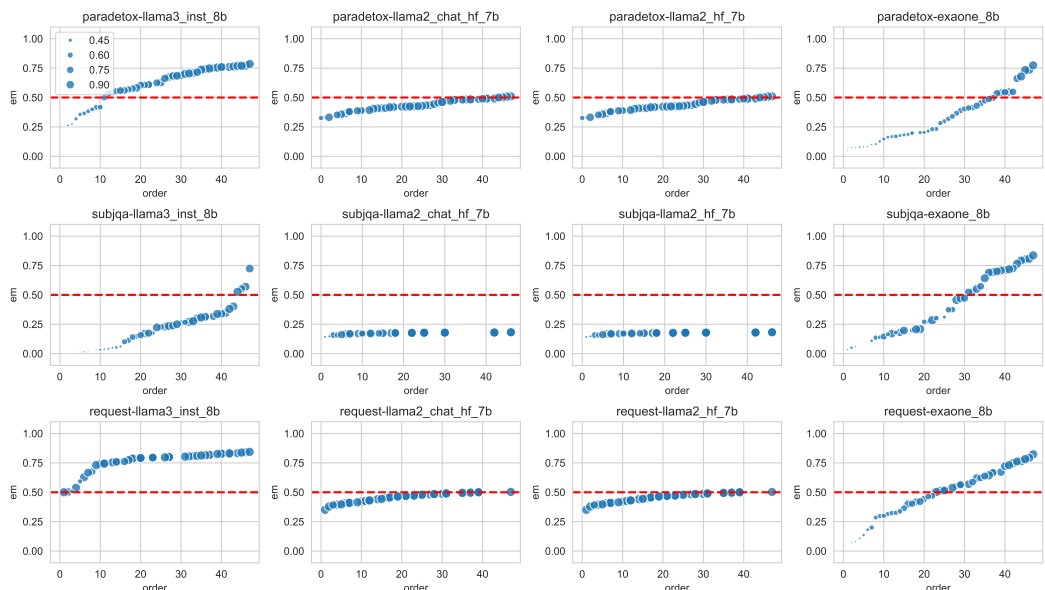

Figure 10: Symbol Choices. Y-axis is the exact match score and X-axis is the sorted index of symbols. The size indicates the proportion of samples whose output matches the binary output format (either `yes` or `no`).

Figure 10 shows the exact match of generation and target labels for 50 symbols (yes and no) for 1,000 train samples. We observed that the performance highly depends on the choice of a symbol. We choose the best symbol for each model and data.

# B  DETAILS ON METHODS

## B.1  STEERING EVALUATION MATRIX

Given $3 \times 3$ matrix $S$ indicating the number of samples for the transition of labels from base generation $B$ to steered generation $S$, each notation $S_{B \to S}$ could be interpreted as the as follows:

1. $S_{n \to y}$ : The original label `no` is successfully steered to `yes`.
2. $S_{n \to n}$ : The original label `no` remains unchanged as `no`.
3. $S_{n \to *}$ : The original label `no` results in an invalid task output `*`.
4. $S_{* \to y}$ : The original invalid output `*` is successfully steered to `yes`.
5. $S_{* \to n}$ : The original invalid output `*` is incorrectly steered to `no`.
6. $S_{* \to *}$ : The original invalid output `*` remains unchanged as `*`.
7. $S_{y \to y}$ : The original label `yes` remains unchanged as `yes`.
8. $S_{y \to n}$ : The original label `yes` is incorrectly steered to `no`.
9. $S_{y \to *}$ : The original label `yes` results in an invalid task output `*`.

## B.2  SUCCESS RATE AND FAILURE OF STEERING

Figure 11 shows all cases for the nominator and denominator of the measured rate

$$\text{Rate}_{\mathcal{N}}^{\mathcal{T}} = \frac{\sum_{(i,j) \in \mathcal{T}} S_{i,j}}{\sum_{(i,j) \in \mathcal{N}} S_{i,j}} \tag{10}$$

We define the success and failure cases $\mathcal{T}$ of steering as follows:

- **Steering Success Rate** is when the the non-`yes` label is steered to `yes` label. Therefore, we set target set $\mathcal{T}_{\text{succ}} = \{S_{n \to y}, S_{* \to y}\}$

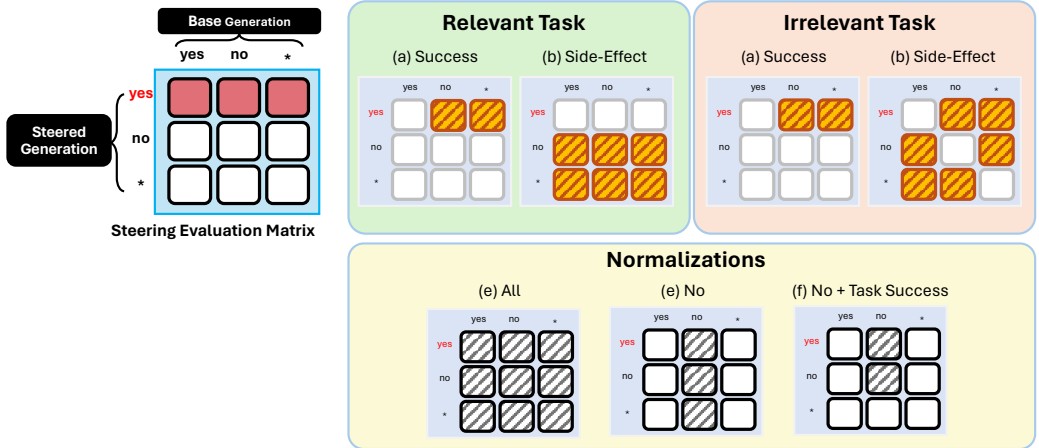

Figure 11: Steering Evaluation Matrix. In the left matrix, the base and steered generations are compared across three possible outcomes: `yes`, `no`, and `*`. We applied activation steering towards `yes`(colored red). The top right section outlines the evaluation method for steering across relevant and irrelevant tasks. In relevant tasks, success indicates correct steering, while side-effect highlights unintended changes. For irrelevant tasks, success and side-effect are tracked, where side-effect is measured as instances where the model generates an incorrect output. The normalization section presents different evaluation methods, including normalization across all outputs, focusing on `no` outputs, and specifically evaluating cases where a response is correctly formatted from `no` to `yes` or `no`.

- **Steering Failure in the Relevant Task** is when the positive label is not achieved. $\mathcal{T}_{\text{rel.fail}} = \{S_{y \to n}, S_{y \to *}, S_{n \to n}, S_{n \to *}, S_{* \to n}, S_{* \to *}\}$.

- **Side-Effect in the Irrelevant Task** is when the original category is converted to another category; therefore, $\mathcal{T}_{\text{irr.fail}} = \{S_{n \to y}, S_{n \to *}, S_{* \to y}, S_{* \to n}, S_{y \to *}, S_{y \to n}\}$

## B.3 NORMALIZATION CONDITION

Normalization term $\mathcal{N}$ plays a pivotal role in understanding the proportion of datasets changed. For failure cases, we can consider all such instances, as they provide insight into where the steering process fails to maintain the correct output format or generates unintended results.

As a result we obtain the following measures

- $\mathcal{T}_{\text{succ}}/\mathcal{N}_{\text{all}}$: Successful steering rate conditional on all samples

- $\mathcal{T}_{\text{succ}}/\mathcal{N}_{\text{neg}}$: Successful steering rate conditional on all samples whose generation was negative in the base generation.

- $\mathcal{T}_{\text{succ}}/\mathcal{N}_{\text{neg+task}}$: Successful steering rate conditional on all samples whose generation was negative in the base generation and preserved format in the steered generation.

- $\mathcal{T}_{\text{fail.rel}}/\mathcal{N}_{\text{all}}$: Failure rate for relevant task conditional on all samples.

- $\mathcal{T}_{\text{fail.irr}}/\mathcal{N}_{\text{all}}$: Failure rate for irrelevant task conditional on all samples.