# OpenReview forum: "Measuring Effects of Steered Representation in Large Language Models"
_ICLR.cc/2025/Conference — ICLR 2025 Conference Withdrawn Submission_

### Official Review · Reviewer_3mPC · 2024-10-28

**Soundness:** 2
**Presentation:** 2
**Contribution:** 2
**Rating:** 3
**Confidence:** 3

**Summary:**

This work looks at the problem of activation steering, which is notion of modifying hidden activations in order to guide the model towards a desired output. The work specifically looks at the direct effects of steering such activations on the consequence generation, which is the part of the generation that follows the steering. The authors develop a framework for this evaluation through the use of counterfactuals, showing how and when steering can have specific effects on the output.

**Strengths:**

The problem itself is relevant, particularly from the perspective of interpretability and faithfulness research. Being able to create direct links between how we prompt models and how they act can have particularly useful outcomes especially as these models continuously grow in scope.

**Weaknesses:**

The work, despite the problem it investigates, does not feel like it is complete. The authors present some interesting results, but the actual contribution here is not very clear and it appears to be quite minimal since the primary difference with prior works is simply using some additional parameters to adjust intermediate hidden activations. Though the method is explained, there's a lack of clarity as to what problem its explicitly attempting to solve and there isn't enough of an exploration of different settings. Furthermore, evaluation is done solely through a lens of a self-defined metric, therefore there may be some direct biasing of the method towards these.

Furthermore, sections 6.3 and 6.4 have been explored quite a bit in the past, such as through needle-in-the-haystack style tasks or simply other long-context problems. The particular link with steered representations here is not presented in a convincing enough way and therefore comes off as nothing more than simply the result of similar observations as those made in prior works with long-context reasoning.

**Questions:**

- I suggest the authors test their steering method on open-ended settings; if it is robust I believe that they should be able to arbitrarily steer the model to any given set of possible labels rather than simply binary pairs.
- I am skeptical that the counterfactual framework proposed is truly measuring the effects of the steering explicitly as there doesn't appear to be any specific controls put in place and therefore measuring there may be a correlation/causation issue here.

---

### Official Review · Reviewer_9syU · 2024-10-31

**Soundness:** 2
**Presentation:** 2
**Contribution:** 1
**Rating:** 3
**Confidence:** 4

**Summary:**

Recently, many representation steering methods have been proposed to enhance alignment and task-specific performance. However, existing evaluations primarily rely on performance metrics, lacking a comprehensive framework for assessing their broader effects. To address this, this paper introduces a counterfactual-based steering evaluation framework that compares base and steered outputs, evaluating both steering success and potential side effects. The proposed framework uncovers model- and task-specific tendencies, including format preservation, query success rates, steering position effects, and impacts on irrelevant tasks.

**Strengths:**

- This paper presents a novel evaluation framework for representation steering, studied independently from task-specific performance, which has not been done before.
- The proposed framework captures critical aspects of representation steering, such as format preservation and the effect of unrelated tasks.

**Weaknesses:**

- The experiments are limited to a few tasks and models. Adding more tasks would greatly strengthen the paper. Additionally, although many representation steering methods have been proposed, this paper evaluates only one specific type. While the experiments appear valid, the limited experimental settings raise questions about the generalizability of the findings produced by the proposed framework.
- The motivation for studying steering methods in the context of in-context learning is somewhat unclear. Since steering methods are commonly applied outside few-shot settings, it would be helpful to clarify any specific reasons for focusing on in-context abilities.
- Some evaluation aspects lack clarity. For example, providing examples or deeper discussion on what constitutes "side effects" would enhance the interpretability of the results.

**Questions:**

- L268-269: Citations for Llama3-8B-inst and Exaone-8B seem broken.
- Typically, I thought steering methods are applied to the last token of each generation step. Is steering at a single token a standard approach?

---

### Official Review · Reviewer_tNRc · 2024-11-04

**Soundness:** 2
**Presentation:** 2
**Contribution:** 2
**Rating:** 3
**Confidence:** 4

**Summary:**

This paper explores the impact of activation steering on the performance of large language models (LLMs). Activation steering involves modifying hidden representations during the forward pass to guide the output in a desired direction. The authors propose a counterfactual-based steering evaluation framework to compare the output of base and steered generations. The paper evaluates LLMs, including Llama3-8B, Llama2-7B, and Exaone-8B, and diverse datasets.

**Strengths:**

The authors introduce metrics, such as Format-Preserving Rate (FPR), Steering Success Rate (SSR), and Side-Effect Rate (SER), this can be useful for formalizing some of the properties desired in steerability. Given steerability is such a desirable property, having formalized metrics to understand this is meaningful.

The authors find that format preservation improves as steering is applied to higher layers. In contrast, steering to layers below the 0.4 percentile leads to significant format disruption. However, the authors find that "The steering outcome can succeed or fail, regardless of whether the format is preserved", so achieving high performance in one metric does not ensure positive directionality in the other. Furthermore, it appears the output form and the analysis of steering success depends on the normalization criteria -- "That is, the
analysis of steering success can vary depending on the criteria used for normalization."

**Weaknesses:**

Controllability tends to differ across models of different sizes, but this papers fails to include models above 8b or shows that findings generalize across different model sizes. This is important to understand the degree of generalization of the findings and metrics.

Even within two models from the same lab, of similar sizes Llama3-8B, Llama2-7B – it would be helpful to have further clarification from the authors if the key findings generalize across both. As it appears the trends differ across datasets. A clear understanding of what metrics actually hold in generalization across datasets and models during rebuttals is important.

As is, the major weakness of this paper is that it appears the metrics remain sensitive to model, dataset and the normalization criteria used. An understanding of how sensitive steering success is to the choice of normalization would also be helpful. For example, the statement " The normalization term can be chosen for the condition-specific measures." suggests that one needs to tailor the normalization -- which would require signficant hyperparameter turning and limit utility of the measures as generalizable.

**Questions:**

How do these metrics generalize to larger models?
Clarify to the reviewers what metrics of those proposed produce consistent trends across all models and datasets (when using the same normalization term across all settings).

---

### Official Review · Reviewer_7KLj · 2024-11-10

**Soundness:** 2
**Presentation:** 2
**Contribution:** 2
**Rating:** 3
**Confidence:** 4

**Summary:**

This paper presents an evaluation framework for activation steering in language models, a technique that modifies the models’ hidden representations by adding task-specific vectors towards a certain direction. Activation steering is used to control the model generation with desired properties by manipulating the hidden states. The paper studies the effect of this particular way of influencing model generation on various axes such as whether output format is preserved, the success rate of steering, the side-effect rate on irrelevant tasks, and the position of steering vector injection in the prompt sequence. Experimental results show varying behaviors across different dataset and models.

**Strengths:**

- The experimental design and results are well-presented, highlighting key observations effectively.

- The paper proposes a robust evaluation framework for investigating the activation steering effect across various dimensions, encompassing both steering-related and unrelated dimensions.

- The research explores the impact of steering position within a sequence, providing valuable insights into the technique and its implications.

**Weaknesses:**

- The novelty of the paper may be limited. The paper conducts an analysis on LM representation steering and presents the observations. Although the comprehensive analysis results can be practically useful to guide the usage of representation steering, the contribution might be slim.

- The paper presentation could become clearer by providing more background information of how activation steering is typically used (e.g. in the beginning sections). The paper studies a particular type of hidden representation modification, and it is better to contextualize the technique along with its variations.

- There are many vague expressions that are not clearly defined and underspecified experimental choices, which affects the overall research clarity. For example, in the end of Introduction, “increase the concept of output” is confusing. For some other examples, see questions below.

**Questions:**

1. What does “increase the concept of output” exactly mean in line 073? Providing more background and context of activation steering in the introduction could help.

2. In Figure 1 bottom panel, the order of “steered output” and “additional prompt” is confusing. The prompt is not input after the output.

3. In Figure 2, what are “in-context knowledge” and “parametric knowledge”? While I understand what they mean, what is the purpose of drawing them in the illustration of the evaluation prompt?

4. Also in Figure 2, why are the activations on the same layer as the steered representations not affected by the steering, indicated by the darker color?

5. What is the reason behind using [S] and [A] tokens introduced in line 131? If they are special tokens for some LLMs, or from other papers on activation steering, please explain.

6. Typo: line 207, in the set for $\mathrm{rel.fail}$, the second term $S_{y\rightarrow y}$ should be $S_{y\rightarrow *}$.

---

### Note · Authors · 2024-11-25

**Comment:**

We thank the reviewers for their constructive feedback.

Based on the comments received, we have decided to withdraw the submission for the following reasons:

1. Evaluation: The dataset and model need to be expanded further to ensure the assessment's robustness and extensibility.
2. Steering Metrics: Improvement is required in the metrics used to assess the influence of steering clearly.

We believe the current state of the submission does not sufficiently meet the standards we aim for.
Once again, we sincerely thank the reviewers for their valuable insights.

**Withdrawal Confirmation:**

I have read and agree with the venue's withdrawal policy on behalf of myself and my co-authors.